

# Coexistence of coarsening and mean field relaxation in the long-range Ising chain

Federico Corberi[1,*], Alessandro Iannone[2†], Manoj Kumar[3‡],
Eugenio Lippiello[4∘] and Paolo Politi[5,6§]

**1** Dipartimento di Fisica "E. R. Caianiello", and INFN, Gruppo Collegato di Salerno,
and CNISM, Unita' di Salerno, Universita' di Salerno,
via Giovanni Paolo II 132, 84084 Fisciano (SA), Italy.
**2** Dipartimento di Fisica E. Fermi, Università di Pisa, Largo B. Pontecorvo 3, 56127 Pisa, Italy
**3** Centre for Fluid and Complex Systems, Coventry University, CV1 5FB, United Kingdom
**4** Dipartimento di Matematica e Fisica, Università della Campania "L. Vanvitelli",
Viale Lincoln 5, 81100, Caserta, Italy
**5** Istituto dei Sistemi Complessi, Consiglio Nazionale delle Ricerche,
Via Madonna del Piano 10, 50019 Sesto Fiorentino, Italy
**6** INFN Sezione di Firenze, via G. Sansone 1, 50019 Sesto Fiorentino, Italy

* corberi@sa.infn.it, † alessandro.iannone93@gmail.com, ‡ manojkmr8788@gmail.com,
∘ eugenio.lippiello@unicampania.it, § paolo.politi@cnr.it

## Abstract

We study the kinetics after a low temperature quench of the one-dimensional Ising model with long range interactions between spins at distance $r$ decaying as $r^{-\alpha}$. For $\alpha = 0$, i.e. mean field, all spins evolve coherently quickly driving the system towards a magnetised state. In the weak long range regime with $\alpha > 1$ there is a coarsening behaviour with competing domains of opposite sign without development of magnetisation. For strong long range, i.e. $0 < \alpha < 1$, we show that the system shows both features, with probability $P_\alpha(N)$ of having the latter one, with the different limiting behaviours $\lim_{N\to\infty} P_\alpha(N) = 0$ (at fixed $\alpha < 1$) and $\lim_{\alpha\to 1} P_\alpha(N) = 1$ (at fixed finite $N$). We discuss how this behaviour is a manifestation of an underlying dynamical scaling symmetry due to the presence of a single characteristic time $\tau_\alpha(N) \sim N^\alpha$.



# 1 Introduction

Systems with long range interactions characterised by an algebraic coupling of the form $r^{-\alpha}$, where $r$ is the distance, are of paramount importance in a variety of situations, ranging from thermodynamics and statistical mechanics [1–3] to astrophysics [4], from hydrodynamics [5] to plasma physics [6] to atomic [7] and nuclear [8] physics, geophysics [9] and many others [1]. According to the value of $\alpha$ fundamental properties of these systems change profoundly. In particular, for large $\alpha$ one usually recovers the features of systems with short range interactions. Lowering $\alpha$, a clearcut distinction must be done between the cases with $\alpha > d$, the spatial dimension, and $\alpha < d$. In the former case some new feature, depending on the specific system at hand, may be determined by the extended interaction with respect to the corresponding short range system. However gross qualitative features are generally not overturned, because the basic assumptions of statistical mechanics, which mostly rely on the additivity property, are retained. This is sometimes called the weak long-range (WLR) regime [10]. For $\alpha < d$, instead, extensivity and additivity are lost. This has important physical consequences since it may lead to [1] non-convex thermodynamic potentials, ensemble inequivalence, negative susceptibilities and non-equilibrium stationary states with ergodicity breaking [11]. Because long range interactions change so much the properties of the system in this case, when $\alpha < d$ one usually speaks of strong long-range (SLR) regime.

In this paper we study the non equilibrium properties of a paradigmatic system of statistical mechanics, a ferromagnet, with SLR interactions. The non equilibrium process we consider is a deep temperature quench. Despite the relevance of the subject, to the best of our knowledge the evolution of such a system has never been studied. Specifically, we will consider an Ising model. As a first attempt to understand this topic, we focus on the one dimensional system, which is more suited for analytical approaches. Indeed, $d = 1$ is the only case where with nearest neighbor (nn) interaction the kinetics is amenable of an exact solution [12], and an analytic framework for the case $\alpha > 1$ was also provided in [13, 14]. Besides that, numerical simulations are obviously less demanding in $d = 1$.

Previous studies [13–16] have shown that with WLR interactions the relaxation phenomenology of the model is akin to the longly studied nn case. Once quenched, after a microscopic time, spin domains of opposite sign form, grow and compete: the phenomenon of coarsening. Let us stress that in this dynamical state there is basically no development of magnetisation, due to the opposite sign of the domains. For $1 < \alpha < 2$, where there is a finite critical temperature $T_c$, such phase-ordering continues forever if $T < T_c$ and the thermodynamic limit is considered: the equilibration time diverges. A finite system, instead, equilibrates in a time which is finite but huge if the size is big, because thermalisation only happens when the domain size has grown comparable to that of the system. For $\alpha > 2$ instead, since $T_c = 0$, coarsening is interrupted even in an infinite system by the onset of equilibration. However the time when this happens grows exponentially as $T \to 0$ so that, for sufficiently low temperature, phase-ordering is promoted to the rank of a macroscopic phenomenon. Of course, apart from such qualitative similarities, there are some quantitative differences between WLR and short range.

For instance, referring to nonconserved dynamics which proceeds through single spin flips, with nn the typical domain size grows algebraically as $L(t) \sim t^{1/2}$ [12] whereas for WLR this happens to be true only for $\alpha > 2$, while there is a non-universal $\alpha$-dependent exponent for $1 < \alpha \leq 2$. Nontrivial low temperature regimes also appear [13,14] and similar differences as $\alpha$ changes are observed in the aging properties [15,16].

On the other side there is the mean field case, $\alpha = 0$. The kinetics of this model is radically different from the one described above. The tiny magnetisation of the initial state, which is of order $1/\sqrt{N}$ in a random configuration, exponentially grows up in a sample, breaking the up-down symmetry and preventing the formation of opposite domains. In this case coarsening is totally absent, the relaxation is trivial, and the system approaches to the low temperature equilibrium state in a time of order one.

The SLR considered in this article exhibits a non trivial scenario which, in some sense, accommodate the two contrasting behaviours discussed above. For a given system size $N$ the non equilibrium ensemble contains a fraction $P_\alpha(N)$ of realisations which display coarsening, the remaining ones behaving similarly to mean field. The choice between the two options is made by each sample of the ensemble basing on the features of the initial state and on the very early stochastic history. A notable consequence of that is the unusual fact that taking the ensemble average one mixes the two kind of dynamics, which are radically different. This makes the usual averaging procedure of non equilibrium statistical mechanics questionable in this case. Let us clarify this with an example. Suppose we have two copies of the system such that one is coarsening and the other is mean field like. The latter will quickly equilibrate developing magnetisation, things which do not happen in the second. Taking the average between the two does not provide a good description of either of them. Related to that, the self-averaging property is also spoiled, i.e. even for large $N$, spatial averages do not correspond to statistical ones.

The dependence of $P_\alpha(N)$ on the size $N$ and on $\alpha$ is also nontrivial. For $\alpha \geq 1$, $P_\alpha(N)$ monotonically converges to 1. This means that coarsening always occurs in a large system. Instead, for a given $\alpha < 1$, $P_\alpha(N)$ is non monotonic in $N$; it initially increases and then decreases to zero for $N \to \infty$ when $\alpha < 1$. Then in the thermodynamic limit all the copies of the ensemble behave akin to mean field. However concluding that mean field is the physically relevant behaviour of a thermodynamically extended system is rash, due to the $\alpha$ dependence. Indeed, the value $N = N^{MF}$ after which $P_\alpha(N)$ starts to decrease diverges as $\alpha \to 1$. Hence a finite system, no matter how large, shows coarsening in some of his instances if $\alpha$ is sufficiently close to 1.

The quantity $P_\alpha(N)$ discussed insofar informs us about the probability that a given realisation will contain domains in its evolution, regardless of the time when these domains will be present. It can be promoted to a time-dependent quantity $P_\alpha(t, N)$, namely the probability that, by looking at a given realisation at time $t$, one finds domains (we use the same symbol for simplicity, since the two quantities have similar meaning. This does not generate confusion). If one computes $P_\alpha(t, N)$ along the whole thermal history, from the quench instant down to the eventual equilibration, one observes that, for large but finite $N$, $P_\alpha(t, N)$ keeps decreasing in time. This is expected because domains in the coarsening samples eventually disappear due to equilibration. We show that the dependence of $P_\alpha(t, N)$ on $\alpha$, $N$ and $t$ is governed by a scaling form similar to the usual ones characterising second order phase-transitions. This suggests that the non equilibrium relaxation of a SLR magnet, although so peculiar, is a dynamical critical phenomenon, as the nn or WLR cases are.

This paper is organised into five sections. The next one is devoted to the definition of the model and of some quantities that will be considered further on. Sec. 3 contains a discussion of the behaviour of the model, based on an independent spin approximation, which is suited to describe the system in the $N \to \infty$ limit. The case of a finite system is considered in Sec. 4

where we first address some properties of isolated domains (Sec. 4.1) and then those of the whole system where many of such can be found (Sec. 4.2). Finally, in Sec. 5 we discuss the results, draw some conclusions and point out open issues.

## 2  Model and non-equilibrium protocol

We consider the one-dimensional Ising model comprising $N$ spins, whose Hamiltonian reads

$$\mathcal{H} = -\sum_i s_i h_i, \tag{1}$$

where

$$h_i \equiv \sum_{j \neq i} J_{ij} s_j \tag{2}$$

is the local field. The model is equipped with a decaying interaction

$$J_{ij} = K(N) r_{ij}^{-\alpha}, \tag{3}$$

where $r_{ij}$ is the distance between two spins $s_i, s_j = \pm 1$ on the sites $i, j$ of a lattice and the Kac factor $K(N) = 1 / \sum_{j \neq i} r_{ij}^{-\alpha}$ is the regularisation necessary to make the energy an extensive quantity [1]. More precisely, $K(N)$ is defined is such a way that $\sum_{j \neq i} J_{ij} = 1$. The distance is evaluated to take into account the periodic boundary conditions, $r_{ij} = \min\{|i-j|, N-|i-j|\}$, The case with $J_{ij} = \delta_{i\pm 1, j}$ is the usual nn situation which corresponds to $\alpha \to \infty$.

It is clear that the Kac normalization tames the divergence of the sum in Eq. (1) in the thermodynamic limit $N \to \infty$ and makes the energy an extensive quantity also in this case. However this does not fix the problem of non-additivity, as one can easily get convinced by controlling that the system obtained by splitting the sample into two parts and bringing them at infinite distance does not have the same energy of the original one.

The equilibrium properties of the model are well known [17–22]. Long-range order is absent at any finite temperature for $\alpha > 2$, while there is a second-order phase transition for $\alpha < 2$. Right at $\alpha = 2$ there is a Kosterlitz-Thouless phase transition with a jump of the magnetisation. For $\alpha = 0$ one has mean field, and mean field critical exponents remain unchanged up to $\alpha = 3/2$.

We consider the evolution without conservation of the order parameter where single spins $s_i$ are randomly chosen and flipped with a transition rate $w(s_i)$ obeying detailed balance, namely $w(s_i)/w(-s_i) = e^{-\beta(E_f - E_i)}$, where $E_i$ and $E_f$ are the energies of the system before and after the elementary move and $\beta$ is the inverse temperature, $\beta = 1/(k_B T)$. We will set the Boltzmann constant to unity in the following. Time will be measured in units of Monte Carlo steps (i.e. $N$ attempted spin flips). Detailed balance leaves freedom in the choice of the transition rates. Imposing the constraint $w(s_i) + w(-s_i) = 1$ leads to the Glauber ones

$$w(s_i) = \frac{1}{1 + e^{\beta(E_f - E_i)}} = \frac{1}{2} \left[1 - s_i \tanh(\beta h_i)\right]. \tag{4}$$

Throughout this paper we will consider the non equilibrium protocol of the quench, where a system is prepared in an equilibrium state at the initial temperature $T_i$ and then instantly cooled to a lower one $T$. In the following we will always consider $T_i = \infty$, where spins are random and uncorrelated, and $T = 0$. Notice that, with the transition rates (4), the evolution at zero temperature proceeds as follows: a spin is randomly chosen and it is flipped if it is antiparallel to the local field. Modifications induced by a finite final quench temperature will be briefly discussed in Sec. 5.

Defining the spatial average $\overline{x}_i = N^{-1} \sum_{i=1}^{N} x_i$ of a quantity defined on site $i$, computed on a given single realisation of the system, the magnetisation density reads

$$m(t) = \overline{s_i} = \frac{1}{N} \sum_i s_i, \tag{5}$$

which varies from sample to sample. Similarly, in the following we will be interested in the spatial average of the local field $\overline{h}_i$. From these quantities one obtains sample independent observables after computing their non equilibrium average $\langle \ldots \rangle$, which is taken over initial conditions and thermal histories. However, as anticipated in Sec. 1, such averaging procedure is not very informative in the case considered here.

## 3 Dynamical process in the thermodynamic limit

In statistical mechanics one is usually interested in the thermodynamic limit $N \to \infty$. However, when a system is brought out of equilibrium the large time sector $t \to \infty$ is also relevant. For the system at hand the order in which the two limits are taken matters, as we will discuss further on. In this section we study the evolution of the model when the thermodynamic limit is taken at the onset. The kinetics of a large but finite system will be studied, also with the help of numerical simulations, in the next section 4.

Because of the quenching from infinite temperature the initial configuration is completely random: spins at different sites are uncorrelated and the spin is uncorrelated to the local field acting on it. We expect the hypothesis of uncorrelated spins continues to be a reasonable approximation at early times. For this reason we will evaluate the zero temperature time evolution of the magnetization neglecting correlations, testing this hypothesis a posteriori.

The magnetization $m$ varies by $2/N$ if we flip a negative spin subjected to a positive field, and by $-2/N$, if we choose a positive spin with a negative field. Within the uncorrelation hypothesis each possibility is the product of independent events. Since a spin is positive/negative with probability $p = (1 \pm m)/2$, if $c_+$ ($c_-$) is the probability that the local field is positive (negative), we can combine the two types of spin flip and obtain

$$dm = \left( \frac{1-m}{2} c_+ - \frac{1+m}{2} c_- \right) \frac{2}{N}. \tag{6}$$

Since the time step related to the random choice of a spin is proportional to $dt = 1/N$ we finally get the following differential equation for the time evolution of the magnetization,

$$\dot{m} = (c_+ - c_-) - m. \tag{7}$$

In order to evaluate $c_\pm$, we observe that with uncorrelated spins, after Eq. (2) the local fields are Gaussian variables with mean $\overline{h}_i = m \sum_r J(r)$, and variance $\sigma^2 = (1-m^2) \sum_r J^2(r)$, where $J(r)$ is the coupling constant $J_{ij}$ between two spins on sites $i$ and $j$ at distance $r = r_{ij}$, previously defined in the first lines of Sec. 2. It is therefore straightforward to write $c_\pm = (1/2)$ $[1 \pm \mathrm{erf}(x)]$, with $x = \left( \frac{\overline{h}_i}{\sqrt{2}\sigma} \right) = m(1-m^2)^{-1/2} S_\alpha(N)$, where $S_\alpha(N) = I_\alpha(N)/\sqrt{I_{2\alpha}(N)}$ and $I_\alpha(N) = \sum_{r=1}^{N/2} r^{-\alpha}$.

Hence Eq. (7) becomes

$$\dot{m}(t) = \mathrm{erf}\left( \frac{m}{\sqrt{1-m^2}} S_\alpha(N) \right) - m. \tag{8}$$

With the limiting behaviours $\mathrm{erf}(x) \simeq \frac{2}{\sqrt{\pi}} x$ for $x \ll 1$ and $\mathrm{erf}(x) \simeq 1 - \exp(-x^2)/(\sqrt{\pi}x)$ for $x \gg 1$ we can approximate Eq. (8) as

$$
\dot{m}(t) \simeq
\begin{cases}
\left( \frac{2}{\sqrt{\pi}} S_\alpha(N) - 1 \right) m \quad , & m < 1/S_\alpha(N) \\[2mm]
1 - m \quad , & m > 1/S_\alpha(N)
\end{cases} .
\tag{9}
$$

These equations show that there is a sudden exponential increase of the magnetisation at early times, $m(t) = m(0) \exp\left[ \left( \frac{2}{\sqrt{\pi}} S_\alpha(N) - 1 \right) t \right]$, followed by a saturation to the equilibrium value, $m(t) = 1 - (1 - m(t_c)) e^{-(t-t_c)}$, where $t_c$ is the crossover time between the two regimes, i.e. the time to attain the crossover value $m = 1/S_\alpha(N)$ through the first exponential regime. The equilibration time $t_\alpha^*(N)$ is given by the sum of $t_c$ and the time to attain saturation through the second regime. Summing up we obtain

$$
t_\alpha^*(N) \simeq \frac{1}{\frac{2}{\sqrt{\pi}} S_\alpha(N) - 1} \ln\left( \frac{1}{m(0) S_\alpha(N)} \right) + \tau_0,
\tag{10}
$$

where $\tau_0 \sim 1$, $m(0) \sim 1/\sqrt{N}$, and $S_\alpha(N) \propto \sqrt{N}$ for $\alpha < 1/2$, $S_\alpha(N) \propto N^{1-\alpha}$ for $1/2 < \alpha < 1$, and $S_\alpha(N) \sim 1$ for $\alpha > 1$.

This equation shows that in the thermodynamic limit $t_\alpha^*(N) \to \tau_0$ for $\alpha < 1$. Therefore in a time scale of order one (when spins can be safely assumed to be uncorrelated) the magnetisation of an infinite system saturates and the model is akin to mean field. Instead for $\alpha > 1$, $t_\alpha^*(N)$ diverges with $N$. At large times the uncorrelation hypothesis is surely not satisfied but the divergence of $t_\alpha^*(N)$ signals that the dynamical behaviour of systems with $\alpha > 1$ is radically different. As a matter of fact we know that for $\alpha > 1$ relaxation is characterised by domain formation and coarsening [13–16].

We also remark that for small, positive $(1-\alpha)$, the first term in Eq. (10) increases for small $N$ and decreases for large $N$. More precisely $t_\alpha^*(N)$ has a maximum at $N_\alpha^* = N_0 e^{1/(1-\alpha)}$, where $N_0$ is an $\alpha$-independent quantity, which diverges exponentially when $\alpha \to 1^-$. This means that approaching $\alpha = 1$ from below we expect a (possibly long) "coarsening" behavior for small $N$ (i.e. $N < N_\alpha^*$), followed by the asymptotic mean-field behavior.

All these scenarios, including the crossover between different dynamical regimes, will be further confirmed by the numerical simulations that will be discussed in the next section.

## 4 Dynamical process for a large finite system

In this section we tackle the problem of the evolution of a large but finite system. We will show that, in this case, formation and coarsening of domains is possible, at variance with the case of an infinite system. In order to do that we start by showing, in Sec. 4.1, that if a sufficiently large domain is formed, it is stable and then the evolution can only proceed by a coarsening process due to the displacement of the domain walls. This study can be conducted analytically in a simple configuration with only two domains where we will also evaluate the time to close one of such, a result that will be later exported to the general quench case in Sec. 4.2. Here we will show that domains are actually formed in a finite system and we will study their evolution.

### 4.1 Stability of individual domains and their evolution

Let us consider a situation with only two domains of size $R \leq N/2$ and $N - R$, respectively, with periodic boundary conditions. We say that a spin is stable at a certain time if it is aligned with its local field and ask the following question: If a spin of the $R-$domain is at distance $X$

from the closest domain wall, is it stable or not? In the short-range models ($\alpha > 1$) only the spin close to the wall ($X = 1$) may be unstable while in the mean-field case ($\alpha = 0$) all spins of the smaller domain $R$ are unstable. We expect to pass from the former to the latter picture when $\alpha$ decreases but how does such transition occur?

Evaluating the interaction of the spin with all others within a continuum approximation and introducing rescaled variables, $x = X/N$ and $r = R/N$, for $\alpha < 1$ we obtain the instability condition

$$x^{1-\alpha} + (r-x)^{1-\alpha} < \frac{1}{2^{1-\alpha}} + \frac{1}{N^{1-\alpha}}. \tag{11}$$

Solving Eq. (11) with the equality sign gives the fraction $x(r)$ of flippable spins in the domain of size $R$. Two special limits are noteworthy: (i) $x$ vanishes when $r = 1/2$ and $N \gg 1$; (ii) all spins are flippable ($x = r/2$) if $r \le r_c = 1/2^{1/(1-\alpha)}$. The latter result shows that sufficiently large domains can be stable in a finite system. Furthermore, for $r_c \le r \le 1/2$, $x(r)$ is a decreasing function of $r$.

Assuming that at each unitary time step all flippable spins do flip, starting from $r(t = 0) = 1/2$ the time $\tau_\alpha(N)$ needed to eliminate the smallest domain can be found from the relation

$$\tau_\alpha(N) = \int_{r_c}^{1/2} \frac{dr}{x(r,N)}. \tag{12}$$

With increasing $N$ the integral in Eq. (12) diverges in $r \to 1/2$, so we can limit ourselves to evaluate such diverging contribution. Close to the upper limit $x$ is small and the $x-$dependence of the second term on the left-hand-side of Eq. (11) is linear, therefore negligible with respect to $x^{1-\alpha}$. Furthermore, if $r = \frac{1}{2} - \epsilon$, at the leading order in $\epsilon$ we obtain $x^{1-\alpha} = (1/N)^{1-\alpha} + c_1(\alpha)\epsilon$ with $c_1 = 2^\alpha(1-\alpha)$. Therefore

$$\tau_\alpha(N) = \int_0 \frac{d\epsilon}{x(\epsilon)} = \frac{1-\alpha}{c_1} \int_{\frac{1}{N}} \frac{dx}{x^{1+\alpha}} = \frac{N^\alpha}{\alpha 2^\alpha}. \tag{13}$$

In order to check this result we have computed $\tau_\alpha(N)$ by means of numerical simulations done on a system with two domains as described above. The results are shown in Fig. 1 and they prove that above picture not only reproduces correctly the exponent, i.e. the dependence on $N$, but also the full $\alpha$ dependence. In fact data agree with great precision with the formula $\tau_\alpha(N) = kN^\alpha/(\alpha 2^\alpha)$, with $k$ of order $1/2$ (best fit to the data, in the time accessed by simulations, provides $k \simeq 0.59$). It is worth noting that such numerical prefactor gives $\tau_1(N) = N/4$, which is what we expect by applying above considerations when only the spin close to the wall may flip, which occurs for $\alpha > 1$.

The above derivation enlightens the mechanism responsible of the domain evaporation which is faster than the ballistic dynamics of domain walls according to which $\tau_\alpha(N) \propto N$. The reason is that the fraction $x(r,N)$ of unstable spins close to the domain walls increases by decreasing $r$, speeding up the dynamics. This however preserves features of domain coarsening. Let us remark that the process becomes ballistic in the limit $\alpha \to 1$. Recalling that the ballistic behavior sets in for $T = 0$ quenches in the WLR regime with any $\alpha > 1$ [14], we conclude that $\tau_\alpha(N)$ crosses over with continuity in passing through $\alpha = 1$.

It should also be stressed that our result $\tau_\alpha(N) \sim N^\alpha$ is found as well at finite temperature for $1 < \alpha \le 2$, both in one dimension [14] and in higher dimension [23–26]. Based on our current understanding this seems a coincidence for a couple of reasons. Firstly, as we will discuss in Sec. 5, the property (13) is spoiled at finite temperatures. Hence the same quantitative result is found in the SLR and WLR cases in different temperature sectors. Secondly, the physical mechanism controlling the closure of the domain is apparently very different in the two cases. With $\alpha < 1$, in a unit time a number of spins is flipped that depends on the

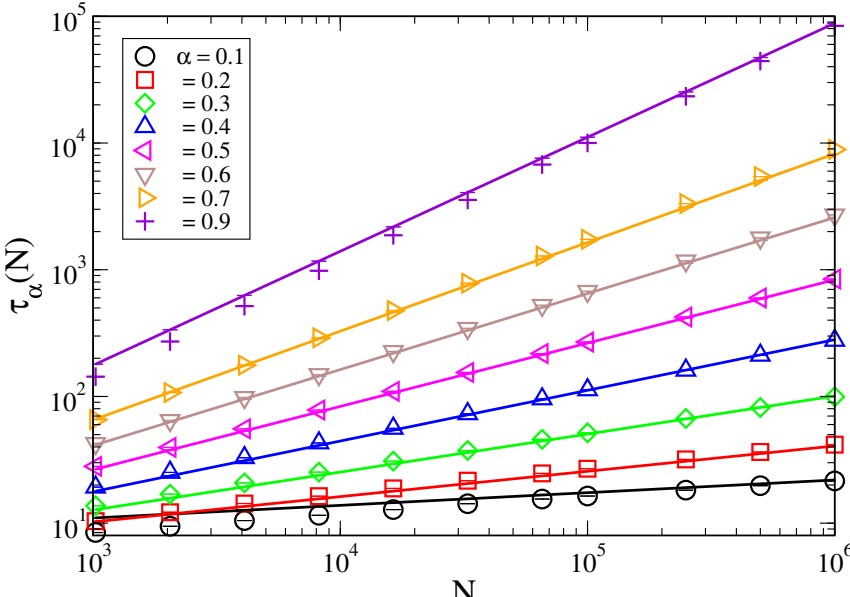

Figure 1: The time $\tau_\alpha(N)$ taken by a system initially made by two domains of size $N/2$ to reach a fully ordered state is plotted against $N$, on a double logarithmic scale. Symbols are outcomes of numerical simulations for various $\alpha$ (see caption). Each curve is averaged over $10^5$ Monte Carlo samples. Continuous lines are the algebraic behavior $\tau_\alpha(N) = k(1/\alpha)(N/2)^a$, with $k = 0.59$.

size of the domain $R$; with $\alpha > 1$, instead, only one interfacial spin can be flipped in a unitary time, but with a probability that depends on $R$ [14]. We also stress that the same dynamical exponent $z = \alpha$ means a dynamics slower than convective (i.e. ballistic) motion if $\alpha > 1$ and *vice versa* if $\alpha < 1$. We will discuss the consequences of Eq. (13) in Sec. 4.2.

## 4.2 Kinetics of a finite system

In the previous section we have discussed the fact that sufficiently large domains, if created, are stable and coarsen. In this section we show that indeed such domains do form in a finite system. Let us anticipate, however, that their development is a stochastic phenomenon which may occur (or not), depending on the different dynamical realisations, with a given probability $P_\alpha$ that we will discuss further below. Before doing this, let us clarify that, from now on, the word *domain* does not refer to spin domains, i.e. regions of the lattice with equally aligned spins, but to local field domains. More precisely, we define a domain as a region of the lattice where $h_i$ does not change sign. Using local field domains is more physical at finite temperature because it neglects fast fluctuations of individual spins. In order to clarify this let us suppose to have a large spin domain one of which quickly flips back and forth. Counting spin domains one has to admit that one domain has split into two. However this has more to do with a random fluctuation rather than with the formation of a new domain. Instead field domains overlook such fluctuations, because the flipping of a single spin does not change much the local fields. At zero temperature field and spin domains almost coincide because spins align in a time of order one with their local field. However, due to the sequential nature of the Monte Carlo evolution, individual spins can temporarily (for microscopic times of order one) remain anti-aligned with the field, introducing a spurious effect similar to the one previously discussed regarding thermal fluctuations.

With this definition, for a given realisation of the process at a generic time $t$ we define the number of domains $D_\alpha(t,N)$ in a system of size $N$ as

$$D_\alpha(t,N) = \frac{1}{2}\sum_i (1-\text{sign}(h_i h_{i+1})).\tag{14}$$

Assuming periodic boundary conditions this number is even by definition. Notice that a configuration with the same sign for all the $h_i$ is referred to as without domains, $D_\alpha = 0$. The probability that, observing a specific sample of size $N$ at time $t$, it is found in a configuration with domains is

$$P_\alpha(t,N) = 1 - \langle \delta_{D_\alpha(t,N),0} \rangle,\tag{15}$$

where $\delta$ is the Kronecker function. If $\delta_{D_\alpha(t,N),0} = 1$ the local field has a constant sign, there are no domains, and the systems behaves qualitatively as a mean field one. Otherwise it contains domains and coarsens.

$P_\alpha(t,N)$ is computed by means of numerical simulations and it is shown in the left panel of Fig. 2 for $\alpha = 0.7$. Different values of $\alpha$ behave similarly and will be discussed in a while. In this figure one sees that, for the chosen value of $N$, $P_\alpha(t,N)$ is definitely finite, despite decreasing in time. The decrease is expected because during coarsening domains are progressively removed until at some time even the two remaining ones are eliminated. Hence one can conclude that a fraction $P_\alpha$ of the dynamical histories develop domains. As it can be seen, their formation occurs immediately after the quench, since $P_\alpha(t,N)$ appears to decrease in time from the very onset of the process. In order to study how such initial formation is influenced by the system size we computed $P_\alpha(1,N)$ whose behaviour, for different values of $\alpha$ is plotted in the right panel of Fig. 2.

Here one sees that $P_\alpha(1,N)$ is a non-monotonic function of $N$. For small sizes it initially increases, reaches a maximum at a certain value $N = N_\alpha^{MF}$ and then decreases to zero. For $N \gg N_\alpha^{MF}$, therefore, the system is found in a mean field like configuration from the very early times basically in all the realisations, which explains the use of the symbol $N_\alpha^{MF}$. The large-$N$ decreasing behaviour of $P_\alpha(1,N)$ is expected after Sec. 3, as in the large-$N$ limit there are no domains, notice however that such decrease is quite slow. The initial increase, instead, shows that not only configurations with domains occur, but also that their probability is enhanced increasing the size up to $N_\alpha^{MF}$.

The dependence of $N_\alpha^{MF}$ on $\alpha$ can be appreciated in the inset of the figure, showing that this quantity quickly increases when $\alpha \to 1$. Data seem to suggest that $N_\alpha^{MF}$ diverges algebraically, $N_\alpha^{MF} \simeq (1-\alpha)^{-n}$ with $n \simeq 4$. Let us also recall that the analogous quantity $N_\alpha^*$, see below Eq. (10), diverges exponentially in the same limit. This discrepancy may be due either to the uncorrelation hypothesis leading to Eq. (10) or to the difficulty to probe numerically the limit of vanishing $(1-\alpha)$. However the key result is that $N_\alpha^{MF}$ diverges, implying that for $\alpha \lesssim 1$ coarsening configurations are by far more probable even in systems of huge size. Notice also that the data suggest that $\lim_{N\to\infty}\lim_{\alpha\to 1^-} P_\alpha(t=1,N) = 1$, meaning that when the limits are taken in this order (but not in the opposite one) the coarsening state is the typical one.

We finally comment on the fact that the behaviour discussed above is just a piece of information of a more general scaling symmetry obeyed by the system, which is reflected by the functional form of $P_\alpha(t,N)$. In order to speculate on this we consider first, as a guideline, the known behaviour of the system with $\alpha > 1$. In this case all the configurations are initially characterised by domains, i.e $P_\alpha(t \simeq 0,N) = 1$, independently of $N$. Coarsening of domains occurs with the growth law $L(t) \sim t^{1/z}$ with $z = 2$ for $\alpha > 2$ and $z = \alpha$ for $1 < \alpha \leq 2$. Domains disappear and equilibrium is reached when $L(t) \sim N$, which occurs at the typical time $\tau_\alpha(N) \sim N^z$, after which one has $P_\alpha(t > \tau_\alpha(N),N) \simeq 0$. In this case, therefore one has the scaling form

$$P_\alpha(t,N) \simeq N^{-b} f_\alpha\left(\frac{t}{\tau_\alpha(N)}\right),\tag{16}$$

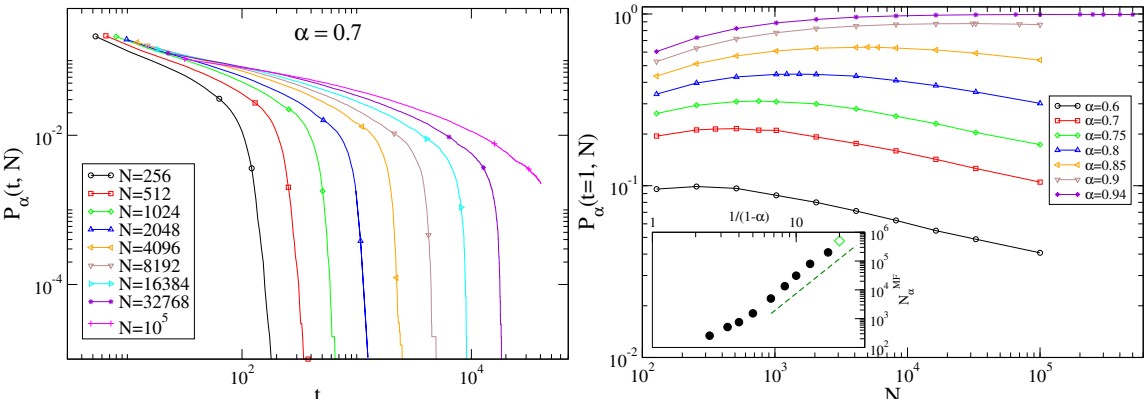

Figure 2: Left panel: $P_\alpha(t,N)$ is plotted against $t$ for $\alpha = 0.7$ and various system sizes $N$ on a double logarithmic scale. Each curve is averaged over $10^5$ realisations. Right panel: $P_\alpha(t=1,N)$ is plotted with log-log scale against $N$ for various values of $\alpha$. In the inset the value $N_\alpha^{MF}$ corresponding to the maximum of any curve is plotted against $1/(1-\alpha)$ with log-log scale. The last point (green diamond) is a lower bound since the maximum of the corresponding curve in the main figure is not yet reached at the longest simulated time. The dashed green line is the behavior $(1-\alpha)^{-4}$.

with $\tau_\alpha(N) \propto N^z$ and $b = 0$. The latter result is due to the fact that for $t \ll \tau_\alpha(N)$ one has $P_\alpha = 1$ independently of $N$.

We maintain now that a similar behaviour is present also for $\alpha < 1$. In this case, the analysis carried out in the last part of Sec. 4.1 suggests that $\tau_\alpha(N)$ is given by Eq. (13). For the exponent $b$, instead, we cannot invoke the same argument leading to $b = 0$ as for $\alpha > 1$, because the constraint $P_\alpha(t \ll \tau_\alpha(N),N) = 1$ does not apply. Rather, we can assume that at very short times $P_\alpha(t \ll \tau_\alpha(N),N)$ is related to the initial, random configuration of spins which produces, see Sec. 3, a Gaussian distribution of the local fields with average $\bar{h}_i = m$ and standard deviation

$$
\sigma = \sqrt{\sum_r J^2(r)} \simeq \begin{cases} \text{const.}\,, & \alpha > 1 \\ 1/N^{(1-\alpha)}, & 1/2 < \alpha < 1 \\ 1/\sqrt{N}\,, & \alpha < 1/2 \end{cases} . \tag{17}
$$

Therefore, with decreasing $\alpha$ the distribution is narrower and consequently the probability to observe configurations with domains must be smaller, as it is indeed observed in simulations. Remarkably, the simple ansatz $P_\alpha(0,N) \simeq \sigma$ seems to be correct. In fact this conjecture gives

$$
b = \begin{cases} 0 \,, & \alpha > 1 \\ 1 - \alpha\,, & 1/2 < \alpha < 1 \\ 1/2 \,, & \alpha < 1/2 \end{cases} , \tag{18}
$$

which is confirmed by simulations, as we are now going to discuss.

The scaling form (16) is surely correct for $\alpha > 1$, because domains always form. We have tested it in the SLR case by numerical simulations, using $\tau_\alpha(N)$ and the exponent $b$ given in Eqs. (13,18), respectively. To do so we look for data collapse of curves for different sizes by plotting $N^b P_\alpha(t,N)$ against $t/\tau_\alpha(N)$. The result of this procedure is shown in Fig. 3, for different choices of $\alpha$. One observes a remarkable data superposition in any case, with the possible exceptions, depending on $\alpha$, of the small and large sectors of $t/\tau_\alpha(N)$. In these regions, however, curve do collapse (or tend to do so) if one looks at a fixed $t/\tau_\alpha(N)$ and let $N$ increase sufficiently. This is enough to conclude that the lack of superposition is just an effect of

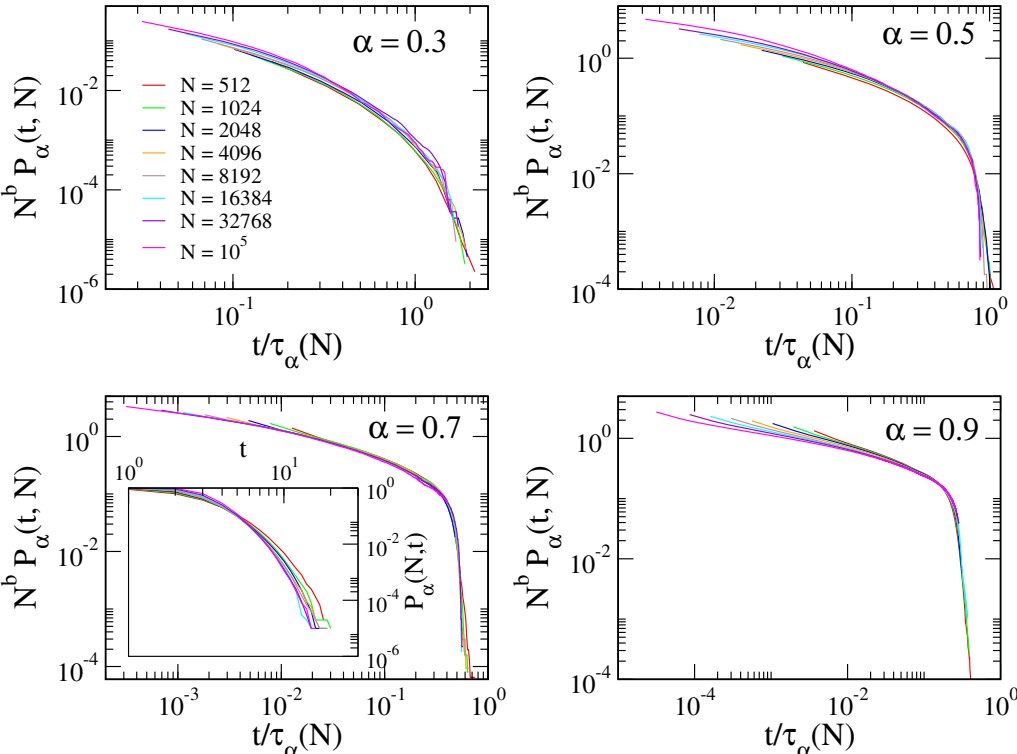

Figure 3: $N^b P_\alpha(t, N)$ is plotted against $t/\tau_\alpha(N)$ on a log-log scale, for different values of $\alpha$ ($\alpha = 0.3, 0.5, 0.7, 0.9$ in the upper left, upper right, lower left and lower right panels, respectively) and various $N$, see keys. $b$ as in Eq. (18) and $\tau_\alpha(N)$ as in Eq. (13). Each curve is averaged over $10^7$ realisations for $\alpha = 0.3$, over $10^6$ for $\alpha = 0.5$, and over $10^5$ for $\alpha = 0.7$ and $\alpha = 0.9$. The inset in the lower left panel reports the (unscaled) data for $\alpha = 0.7$ but in a quench to $T = T_c/2$.

preasymptotic corrections for finite $N$. Corrections at small $t/\tau_\alpha(N)$ are particularly evident for large $\alpha$ because out of this sector scaling is excellent. Such deviations can perhaps be ascribed to the divergence of $N_\alpha^{MF}$ as $\alpha \to 1$. In the large $t/\tau_\alpha(N)$ region corrections probably arise as due to the basically different kinetics in the final stages of the process when even the last few surviving domains are expiring. Notice indeed that this happens when $P_\alpha(t, N)$ is already very small. In conclusion, Fig. 3 strongly support the scaling form (16) with the quantities $\tau_\alpha(N)$ and $b$ given in Eqs. (13,18).

## 5 Conclusions

In this paper we have considered the non equilibrium kinetics of the $1d$ Ising model with long range interactions decaying algebraically with an exponent smaller than the spatial dimension, the so called SLR regime. As compared to the contrasting behaviours of the limiting cases with infinite range (i.e. mean field) and short range interactions, the SLR case shows the coexistence of both of them. Specifically, different realisations of the ensemble are found either in a mean-field like state or in a coarsening one. This can be interpreted as a new instance of a dynamical symmetry breaking phenomenon. With mean field each sample breaks the $Z_2$ symmetry globally building either positive or negative magnetised states. The choice must be

traced back to the properties of the initial condition. With short (and even WLR) interactions the symmetry is broken locally inside the freshly formed post-quench domains whose sign is determined by the initial state and by the early history. With SLR interactions there is space for breaking the symmetry group into a larger set of subgroups, because either a global or a local symmetry breaking occurs, the probability of each being given by $P_\alpha$ which again is determined by the configuration at the quench time and by the early evolution. Such symmetry breaking scenario is further enriched by the role played by the system size, the local symmetry breaking with domains formation being disfavoured upon increasing $N$ above a certain ($\alpha$-dependent) value $N_\alpha^{MF}$.

From a thermodynamic perspective, changes of symmetries correspond to phase transitions. Previous considerations imply, therefore, that a phase transition occurs at $\alpha = 1$ in the $1d$ Ising model. From the point of view of the properties of $P_\alpha$, the transition is of a continuous type for a large but finite system, because we have shown that $\lim_{\alpha \to 1^-} P_\alpha(t \gtrsim 0, N) = 1 = P_{\alpha>1}(t \gtrsim 0, N)$. However it turns into discontinuous in an infinite system, because $\lim_{\alpha \to 1^-} \lim_{N \to \infty} P_\alpha(t \gtrsim 0, N) = 0 \neq \lim_{N \to \infty} P_{\alpha>1}(t \gtrsim 0, N) = 1$.

The rich scenario addressed insofar is limited to zero-temperature quenches. A natural progress would be understanding the effect of finite quenching temperatures $0 < T < T_c$. The question is not trivial because temperature is known to be irrelevant both in the nn case [27–29] and in mean field [13, 14]. Irrelevant means that the overall qualitative behavior is the same for all $T < T_c$ and universal quantities such as exponent do not depend on $T$. Instead, with algebraic interactions and $\alpha > 1$ there is a difference between quenches to $T = 0$ and to $0 < T < T_c$, both in $d = 1$ [13, 14] and in $d = 2$ [24, 30]. The situation in the present $1d$ case with SLR is shown in the inset of Fig. 3. Here we report the behavior of the quantity $P_\alpha(t, N)$, plotted against time for various values of $N$, for the model with $\alpha = 0.7$ quenched to $T = T_c/2$ ($T_c = 2^\alpha J/(1 - \alpha)$ [1]). A similar behavior is found for different values of $\alpha$. In this figure one sees that data for sufficiently large values of $N$ collapse, without need of any rescaling: this is in striking contrast with quenches to $T = 0$, see left panel of Fig. 2. At finite $T$ the small deviations from a perfect collapse are less evident with increasing the size $N$ and are most likely due to finite-size corrections.

This means that temperature changes radically the behaviour of the system. In particular, independence on $N$ is the signature of the mean field behaviour, as witnessed by the fact that $P_\alpha$ goes to zero – hence domains expire – in a microscopic time independent on $N$. We conclude that temperature breaks the up down symmetry and instates the mean field mechanism. Of course, we expect this to happen with a crossover scenario: the smaller is $T$, the later it will kill the domains. Indeed, we have checked that with a very low temperature we observe the same pattern as with $T = 0$ in the time domain accessed by simulations. However, if one expects sufficiently, mean field prevails. $T = T_c/2$ considered in Fig. 3 is evidently a rather high temperature under this respect: one must not wait that long in this case, it happens in a microscopic time of order 5-10.

The material presented in this article is a first study of the post-quench kinetics of the Ising model with SLR interactions. As such, it focuses mostly on basic features of the dynamical state. However, several properties remain yet unexplored among which the precise nature of the initial coarsening regime and in particular the growth law $L(t)$ of the domains size. Indeed, it is known that for $\alpha > 1$ the relation $\tau_\alpha(N) \sim N^\alpha$ (Eq. 13) corresponds to $L(t) \sim t^{1/z}$ with $z = \alpha$. However, given the strong long-range nature of the present case, this matter would require an analysis on much longer timescales than those addressed in this paper.

Besides that, the aging properties, encoded by two-time quantities, as well as the case with $d > 1$ are totally unexplored. Furthermore, given the important role played by symmetries discussed above, ferromagnetic systems with a continuous symmetry are also expected to exhibit peculiar features.

## Acknowledgements

MK would like to acknowledge the support of the Royal Society - SERB Newton International fellowship (NIF\R1\180386). The numerical computations presented here were done on a *Zeus* HPC of Coventry University. EL and PP acknowledge support from project PRIN2017-98CZLJ .

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
