# Peer review of "Coexistence of coarsening and mean field relaxation in the long-range Ising chain"

_SciPost Physics, doi:SciPost Phys. 10, 109 (2021)_

## Round 1 · Referee Report · Anonymous (Referee 1) · 2021-4-11

Report

The authors have made the required changes to the manuscript. I recommend publication of the paper in SciPost Physics.

---

## Round 1 · Referee Report · Anonymous (Referee 3) · 2021-4-19

Report

The authors have addressed my previous comments. I recommend publication in SciPost.

---

## Round 1 · Author Response

With the present cover letter we resubmit the paper “Coexistence of coarsening and mean field relaxation in the long-range Ising chain” .

We thank all the Referees for their general appreciation of our work, and for the useful suggestions and remarks. We have taken into account all the points raised, and changed the manuscript accordingly. Answers to the Requested Changes are reported below. Relevant changes in the manuscript are highlighted in red.

We are confident that, in the present form, the paper is suited for publication on SciPost.

Sincerely

Federico Corberi Alessandro Iannone Manoj Kumar Eugenio Lippiello Paolo Politi

Referee of report 1

  1. Correct the few typos. We have corrected the typos.

Referee of report 2

  1. The scaling of the characteristic length L(t) of the coarsening is mentioned in the abstract but has never been discussed explicitly in the text. At least I do not find any numerical data supporting it. Of course, \tau \sim N^z is shown with z = 1/\alpha. However, one should keep in mind that z = 1/\alpha is not true for all coarsening systems, for example in fluids.

The paper does not contain explicit results on L(t), we agree with the Referee on this. Henceforth, the last sentence in the abstract was inappropriate. We dropped it out and replaced with one concerning the characteristic time \tau_\alpha (N), the object we extensively investigate here. We thank the Referee for pointing this out. As the Referee points out, the relation between L(t) and \tau might not be straightforward for the problem at hand. We have briefly discussed this in a new sentence towards the end of the conclusions.

  1. No unit of time is defined for the numerical results.

We have defined the time units — Montecarlo steps — before Eq.(4).

  1. Eq. (9) is the heart of their theoretical finding. Can the authors comment on its validity if one fixes the initial condition to m(0)=0 ?

Eq.(9) [(10) in the new version] has been derived within a procedure that is mean field in spirit, where therefore all spins are equivalent. In a real system (and in simulations as well) spin flips occurs sequentially. Henceforth, even in the very unlikely case of an initial condition with exactly m(0)=0, after the first spin flip one would have a finite (small) m. Let us stress, however, that the typical absolute value of m(0) is sqrt(N).

  1. In Fig.2 and its caption the probability is written as P (N , t) whereas in the text I see the order of N and t is changed as P (t, N ). It is better to stick to one.

Done.

  1. The pre-asymptotic correction should be vanishing with increase of N. However, having a careful look at the data in Fig.3 (a) reveals the following:
With the reasonable assumption that the N = 10^5 data in Fig. 3(a) is the closest one representing the master curve, one can see that the deviation of the data for N < 10^5 is not systematic. An explanation is required.

Actually a close look to Fig. 3 shows that, for any value of \alpha the corrections are systematic, at least when the noise in the data can be assumed to be negligile. In the less noisy case (\alpha =0.9) this is very clearly appreciated. Data for alpha as small as 0.3 (those to which the Referee makes reference) are very noisy, indeed we had to go to 10^7 realisations to clean the curves as in Fig. 3 (see caption). But also in this case it seems to us that one can conclude that when the data are sufficiently free from noise (i.e. left part of the figure, N not too large) the corrections look systematic (approach from below). Of course noise, when relevant, can mask this.

  1. In the conclusion: Please provide references for the study of the nearest neighbor Ising model regarding universality of the growth exponent at different T .

The relevant theoretical reference was already provided in the paper (Ref. A. Bray, PRB 41, 6724 (1990)). We have added a couple of numerical references (Conclusions).

  1. The author should exclusively comment on the growth exponent for T < Tc in d = 2 ? They must be aware of the seminal analytical works by Bray and Rutenberg in this regard and also if there exists any numerical evidence supporting that.

We have added a short discussion on this point and the relevant references (at pag. 6).

  1. What is the value of Tc for \alpha= 0.7 ? The authors claim that the data for T = Tc /2 presented in the inset of Fig. 3 is N independent for large N which is not so convincing. Is this simply finite-size effects ?

We have explicitly mentioned the value of Tc (Conclusions) and the appropriate reference. Regarding the data in the inset of Fig. 3, we have added a sentence in the Conclusions to discuss the small deviations from perfect collapse. 

Typos have been corrected.

Referee of report 3

1.(refers to item 1. of the Weaknesses section) At the beginning of Sec. II, first, a definition of J_ij is given inline in the text, definition that includes the Kac normalization factor (which maybe deserves a line of comment, absent here) and then Eqs. (1) and (2) are presented. However, in my opinion one should reverse the order of presentation, that is, present Eqs. (1) and (2) before and afterwards promote also the definition of the J_ij to a numbered equation, for the sake of clarity.

Done. The Kac factor, which is standard in the long-range literature, was already briefly discussed. Here we have added a reference to its definition (Ref. [1]).

  1. The sentence "For intermediate values of \alpha, but having \alpha > d, the spatial dimension, some new feature, depending on the specific system at hand, may be determined by the extended interaction, however the basic assumptions of statistical mechanics, which mostly rely on the additivity property, are retained." is not very clear and is very heavy to read: my suggestion is to reformulate it.

Done.

  1. In addition to Ref. [1], the book "Physics of long-range interacting systems" by A. Campa, T. Dauxois, D. Fanelli, and S. Ru]o (Oxford, 2014) should be cited.

Done.

  1. The terminology "weak long range" and "strong long range" is explained at the beginning if the paper. However, being such a terminology not completely agreed upon, the authors should explicitly cite some references where their terminology is introduced and/or discussed.

Done (we have added a reference in the introduction (after defining the acronymous WLR).

Typos have been corrected. References have been expanded.

---

## Round 1 · List of Changes

Changes are highlighted in red in the resubmitted version.

---

## Editorial Decision

published